# Assessing operational readiness: Regulatory landscape and compliance in zimbabwe for medical devices and in vitro diagnostic medical devices

**Charles Chiku**[1]*, **Talkmore Maruta**[2], **Fredrick Mbiba**[3], **Justen Manasa**[4]

1 Regulation and Prequalification Department, World Health Organization, Geneva, Switzerland,
2 Programs, African Society for Laboratory Medicine, Lusaka, Zambia, 3 The Health Research Unit, Biomedical Research and Training Institute, Harare, Zimbabwe, 4 Department of Oncology, Faculty of Medicine and Health Sciences, University of Zimbabwe, Harare, Zimbabwe

* charleschiku@gmail.com

**Data Availability Statement:** Data was deposited in Open Science Framework (OSF) Registries under the Project name Medical devices regulation

## Abstract

The regulation of medical devices and In Vitro Diagnostic (IVD) medical devices have lagged significantly, especially in low- and middle-income countries. Disparities in regulating medical and IVD medical devices in Africa are below the global average. This may translate to poor access to quality-assured medical and IVD devices, resulting in undesirable health outcomes. Operational readiness to regulate medical and IVD devices at the Medicines Control Authority of Zimbabwe (MCAZ) was assessed. The aim was to determine the strengths and gaps and propose an action plan that can be monitored and evaluated to assess progress over time. We used the World Health Organization (WHO) Global Benchmarking Tool for medical devices and IVDs methodology to evaluate regulatory oversight of these products. Purposive sampling was used for data collection using researcher-administered global benchmarking tool factsheets and document reviews to evaluate the implementation of the regulatory functions. The regulatory functions assessed were the National Regulatory System, Registration and Market Authorization, Vigilance, Market Surveillance and Control, Licensing Establishment, Regulatory Inspection, Laboratory Testing, and Clinical Trials Oversight. The MCAZ attained maturity level 1, with a regulatory system score of 79%, registration and market authorization 44%, vigilance 27%, market surveillance and control 40%, licensing establishment 62%, regulatory inspection 68%, laboratory testing 88%, and clinical trials 18%. Condoms and gloves were the only regulated medical devices in Zimbabwe. IVDs were not regulated by the MCAZ. This review showed that the regulatory system is not robust, fit for purpose, responsive, transparent, or proportionate to the risk classification of medical devices and IVDs. It is crucial to amend the Medicines and Allied Substance Control Act to incorporate the definition and classification of medical devices and IVDs, regulatory authority establishment, licensing and registration, quality management system, conformity assessment, post-market surveillance, labeling and instructions for use, capacity building and training, and international harmonization.

in Zimbabwe: an evaluation of operational readiness, doi: https://doi.org/10.17605/OSF.IO/C5JXG.

**Funding:** The author(s) received no specific funding for this work.

## Introduction

Medical devices and In Vitro Diagnostic (IVD) medical devices are essential for the public to reach their highest health standards. Regulations for medicines and vaccines have existed for many years. Yet, regulation of medical devices and IVDs lags. This is especially true in low and middle-income countries. We should recognize that overseeing medical devices and IVDs is not standalone. It should align with regulations for other medical products, such as medicines and vaccines. It should also align with broader government policy goals [1]. Medical devices are any instrument, apparatus, appliance, software, implant, reagent, or other similar or related article intended to diagnose, prevent, monitor, treat, or alleviate disease. It also includes products intended to affect the structure or any physiological process within the human body, provided they do not achieve their principal intended action by pharmacological, immunological, or metabolic means.

On the other hand, an IVD refers to a medical device, whether utilized independently or in conjunction, designed by the manufacturer to examine specimens derived from the human body in vitro, primarily aimed at supplying information for diagnostic, monitoring, or compatibility purposes [2]. Medical device and IVD regulations are laws that govern the design and development, manufacturing, clinical trials, and distribution of medical devices. The aim is to ensure that they are safe and perform as intended by their manufacturers [3, 4]. The primary objective of regulating medical devices is to facilitate access to safe and effective devices with acceptable performance and quality to ensure patient and user safety.

In 2016, the World Health Organization (WHO) conducted a review to evaluate the availability of medical device regulations globally. In the Afro region, 40% of countries have no medical device regulations. Thirty-two per cent have some regulations, and 28% lacked data. Globally, 58% of the WHO members had regulations in place. Zimbabwe was reported to regulate the medical device market placement. However, the data are vague. This gap in medical device regulation is crucial. The African region's regulation lags behind the global average and hence may lead to access to lower-quality medical devices. It may also limit patient access to healthcare technologies. Zimbabwe only had elements of placing medical devices on the market. The country lacked pre-market and post-market regulations. It is unclear whether the framework in Zimbabwe has effectively evolved to regulate medical devices [5]. This gap in medical device regulation between the Afro region and the global average is crucial as it may lead to lower-quality medical devices and limited patient access to healthcare technologies.

A review by Hubner et al. of medical device regulations in the 14 member countries of the College of Surgeons of Central, Eastern, and Southern Africa (COSECA) showed that Zimbabwe had a legal framework for regulating medical devices. However, the regulation was not formal. It covered conformity assessment, import, export, and post-market surveillance for condoms and gloves. However, there is a dearth of medical device literature specific to Zimbabwe [6].

A qualitative study by Dacombe et al., which assessed the regulation of HIV-Self Testing IVDs in Malawi, Zambia, and Zimbabwe, showed that the reference laboratory monitored the quality of HIV-Self Testing IVDs in all three countries. The responsibility to regulate these devices overlapped between the MCAZ and the Medical Laboratory and Clinical Scientists Council of Zimbabwe. Stakeholders indicated a poor understanding of the regulatory process and lacked clarity and coordination roles [7].

The lack of research on Zimbabwe's governance system for medical devices and IVD regulations, as stated by Hubner et al. in the COSECA region, indicated that Zimbabwe has no formal regulatory system for medical devices except for gloves and condoms. The extent to which Zimbabwe's regulatory system governance is an essential driver for implementation is yet to

be adequately appraised because of a lack of studies specific to the Zimbabwean context. Recent studies have shown an absence of medical device regulation literature in the COSECA region [6]. It is unclear whether Zimbabwe's medical device and IVD regulatory framework has evolved and whether it is ready to regulate these devices.

## Methods

A cross-sectional study used the WHO global benchmarking tool for medical devices and IVDs. It examined MCAZ's oversight of medical devices and IVDs from June to August 2022. The researcher interviewed staff members at the MCAZ who were responsible for implementing medical device regulatory functions. We used the WHO global benchmarking tool factsheets for data collection. We wanted to evaluate the quality of implementing medical devices' regulatory functions. Interviews were complemented with a desk review of documents and records. This was done to verify the implementation of the regulatory functions. Indicators and sub-indicators were used to evaluate the implementation of the regulatory functions. **Tables 1** and **2** show the definitions of regulatory functions and indicators that were assessed to determine each regulatory function's implementation quality [8].

### Study setting

MCAZ is the national regulatory authority that regulates medicines and allied substances according to the Medicines and Allied Substances Control Act [Chapter 15.03]. The selection of participants for interviews from the MCAZ's Medical Devices and Microbiology Unit was purposive. The objective was to identify individuals capable of providing valuable insights into the regulatory processes, practices, and challenges within the MCAZ. The criteria for selecting participants for this study were designed to ensure a comprehensive representation of key stakeholders in medical devices and IVDs regulations. The inclusion criteria comprised three main categories: Firstly, officials directly engaged in regulatory decision-making were considered integral to providing insights into the decision-making processes. Secondly, individuals

**Table 1. Definitions of assessed regulatory function.**

| Regulatory Function | Description |
|---|---|
| National Regulatory System | The legal and regulatory framework supports the regulatory system's functions in ensuring the quality, safety, and performance of medical and IVD devices. |
| Registration and Market Authorization | The issuance of marketing authorizations (also referred to as product licensing or registration) happens when medical and IVD devices have met the requirements of standardized conformity assessment. |
| Vigilance | The science and activities related to preventing, detecting, assessing, and understanding adverse effects or other medical product-related problems. This ensures that medical products meet quality, safety, and performance requirements throughout the product lifecycle. |
| Market Surveillance and Control | The process of ensuring ongoing compliance of products on the market with quality, safety, and performance requirements. |
| Licensing Establishment | This function guarantees the quality, safety, and performance of medical products used within or exported out of the country. It does this by licensing establishments involved in medical products' value chain and life cycle. |
| Regulatory Inspection | Auditing establishments throughout the value chain and life cycle of medical devices to ensure compliance with laws, regulations, approved standards, norms, and guidelines. |
| Laboratory Testing | The national regulatory authority verifies the manufacturer's performance claims. This supports pre-market approval or a change to marketing authorization. |
| Clinical Trials Oversight | Refers to the legal mandate of the national regulatory authority. It authorizes, regulates, and, if necessary, terminates clinical trials. |

**Table 2. Assessed indicators to determine the status of regulatory functions.**

| Indicator | Description |
|---|---|
| Legal provisions, regulations, and guidelines. | The presence or absence and implementation of legal provisions, regulations and guidelines for each regulatory function. |
| Organization and governance. | The structure and line of authority among and within all institutions participating in the regulatory system is defined, documented and implemented. |
| Policy and strategic planning | A national medical devices and IVDs policy, aligned with health policy, exists and is implemented. |
| Leadership and crisis management. | National regulatory authorities have the appropriate tools and institutional framework for co-coordinated action to manage crises. Leadership ensures that strategic priorities and objectives are well-known and communicated throughout the national regulatory authority. |
| Transparency, accountability, and communication. | Ensure the implementation of mechanisms to publish, information sharing among stakeholders and promotion of transparency on regulatory applications, including authorized, suspended, rejected, and completed applications. |
| Quality and risk management systems. | The confirmation of the national regulatory authority's implementation of the main principles of a quality management system, which includes the application of risk management principles for all regulatory functions, has been updated to include the required documentation. |
| Regulatory process. | Processes that are implemented to fulfil each regulatory function. |
| Resources (including human, financial, infrastructure, equipment, and information management systems). | Ensure that top management reviews the national regulatory authority's performance, the areas for improvement, and the status of resources at planned intervals. The purpose is to ensure the continued effectiveness of medical products regulatory oversight and alignment with the organization's strategic direction. |
| Monitoring progress and assessing impact | The existence and implementation of requirements and procedures for monitoring, supervising, and reviewing the performance of national regulatory authorities and affiliated institutions. |

with substantial expertise and experience in health product regulation were targeted, ensuring a wealth of knowledge and a nuanced understanding of the regulatory landscape. Lastly, individuals responsible for critical regulatory functions, such as product registration, licensing, quality control, inspections, pharmacovigilance, and post-marketing surveillance, were included to capture diverse perspectives and experiences within the regulatory domain. Through this careful selection process, the study aimed to gather a well-rounded and insightful perspective on health product regulation from those actively shaping and implementing regulatory policies. A total of two participants were interviewed. Additionally, an extensive desk review of pertinent documents was conducted to validate the current status of legal provisions, regulations, and guidelines governing the regulatory landscape. This comprehensive approach ensured a nuanced understanding of the regulatory framework within the MCAZ.

## Data analysis

Descriptive analysis was performed to describe the quality of regulatory function implementation based on the cumulative score of the sub-indicators for each regulatory function. A computerized global benchmarking tool scoring algorithm was used to determine the

implementation status of each indicator for each regulatory function. Scoring was performed to measure the proportion of sub-indicators implemented as the numerator and the total sub-indicators as the denominator.

1. Not implemented: No evidence was provided to demonstrate any degree of implementation of the sub-indicators. This status was assigned where a score of 0% as a percentage is attained.

2. Ongoing implementation: Some actions/steps/activities were undertaken to implement the concerned sub-indicators. However, the sub-indicators still need to be fully implemented. A score of 25% was assigned to this category (a score of 25% as a percentage).

3. Partially implemented: Some actions/activities showed full implementation of the sub-indicator; however, such full performance is recent or relatively new, with little cumulative data for consistent execution. Supporting documented evidence is expected to show the current full implementation of the concerned sub-indicators. For mathematical scoring, 'partially implemented' is scored as 0.75 out of one (i.e., 75% as a percentage).

4. Fully implemented: some actions/activities demonstrate the consistent and full implementation of the sub-indicator over time. Supporting evidence is expected to illustrate the full, consistent implementation of the sub-indicator (i.e., shown over time and through repetition of the process and outcome). 'Fully implemented' is scored as one out of one (i.e., 100% as a percentage).

Maturity Level is a concept adopted from the International Organization for Standardization (ISO) 9004:2018 standard, " *Five levels of organizational maturity for quality management systems have been established to enhance functionality: a lack of a formal approach, a passive approach, a formal and stable system approach, an emphasis on continuous improvement, and the highest level, which is considered the best in the industry*" [9]. The computerized global benchmarking tool algorithm was used to assign maturity levels based on the cumulative scoring of the sub-indicators under that function. The maturity level classification allows for identifying systems that require improvement and advanced ones that can facilitate reliance and greater regulatory cooperation. The maturity of regulatory systems is divided into four levels, characterized as follows.

- Maturity Level 1: Regulatory systems in which some elements of regulatory systems exist correspond to "no formal approach" (ISO 9004:2018).

- Maturity Level 2: evolving national regulatory systems that partially perform essential regulatory functions; corresponds to "reactive approach" (ISO 9004:2018);

- Maturity Level 3: stable, well-functioning and integrated regulatory systems; corresponds to "stable formal system approach" (ISO 9004:2018); and

- Maturity Level 4: regulatory systems operating at an advanced level of performance and continuous improvement correspond to "continual improvement emphasized" (ISO 9004:2018) [10].

**Ethical considerations.** Ethical approval for this study was obtained from the Medical Research Council of Zimbabwe (MRCZ/A/2900). Participation in the study was voluntary. The participants were able to stop the interview at any time without any explanation, if necessary. Written informed consent was obtained from each study participant before each interview. The interview content and interviewees' identities were kept anonymous.

## Results

Generally, the quality of the implementation of regulatory functions was below optimum. Regulatory functions were implemented only for condoms and gloves. The use of other medical devices was not regulated. The Lot Release regulatory function was not assessed because it does not apply to medical devices and IVDs. The quality of implementation of each regulatory function ranged between 18% and 88% for condoms and gloves, respectively. The clinical trial oversight function was the lowest (18%), and Laboratory Testing scored 88%.

The national regulatory system did not have explicit legal provisions and regulations that required medical devices to meet specific safety, quality, and performance requirements throughout their lifecycle. Additionally, the Medicines and Allied Substances Control Act did not define the roles and responsibilities of the institutions involved in the medical device regulatory system. Similarly, there were no legal provisions or relevant regulations to take action on the recall, suspension, withdrawal, and destruction of substandard and falsified medical devices. Eleven documents were reviewed to complement the interviews. (S1 Table: List of documents reviewed).

The regulatory functions assessed were all at a maturity level of 1. Maturity level 1 was the least used to measure each indicator's implementation quality. **Fig 1** shows the maturity level of each regulatory function.

The national regulatory system for medical devices lacks clear legal guidelines that mandate safety, quality, and performance standards for the entire product life cycle.

Medical devices lack legal compliance for registration and authorization. No conformity assessments were proportional to the risk class of medical devices. Draft IVD regulations were pending approval.

The vigilance function was insufficient to prevent, detect, assess, and understand adverse effects and other medical device-related issues throughout the device's lifecycle, except for condoms and gloves.

Efforts are currently underway to address market surveillance and control. Import regulations for medical devices have been drafted, but legal provisions for controlling import activities and market surveillance are unavailable. The MCAZ does not have legal provisions for dealing with substandard and falsified medical devices.

No legal provisions are required for registration for economic operators in the medical device value chain based on compliance with good practices. The MCAZ lacked the power to license medical devices and IVDs value chain establishments, except for condoms and gloves.

No legal provisions or guidelines were required to define the regulatory framework for inspecting and enforcing medical devices, except for condoms and gloves. No updated national good practices, regulations, or norms are mandatory to guide economic operators in their medical device submissions.

Well-established testing for condoms and gloves is in place, but the legal basis is unclear. The steps taken to establish IVD medical device laboratory testing are in progress, but there is no evidence of legal provisions or regulations for reliance on other laboratories' decisions. There were no clinical trial oversight regulations, national regulatory authority notifications, or compliance with good manufacturing practices.

## Discussion

This is the first study of Zimbabwe's medical devices and IVD regulations, focussing on all regulatory functions. The regulatory system concerning medical devices and IVDs in Zimbabwe is notably deficient due to an underdeveloped legislative framework for regulatory functions. The Medicines and Allied Substances Control Act was enacted in 1969 and only focused on

| Function | Sub-indicator results | Percentage of implemented sub-indicators | Maturity level |
|---|---|---|---|
| National Regulatory System | 47.25/60.0 | 79.0 % | 1 |
| Registration And Marketing Authorization | 15.25/35.0 | 44.0 % | 1 |
| Vigilance | 7.0/26.0 | 27.0 % | 1 |
| Market Surveillance and Control | 10.75/27.0 | 40.0 % | 1 |
| Licensing Establishment | 11.75/19.0 | 62.0 % | 1 |
| Regulatory Inspection | 17.75/26.0 | 68.0 % | 1 |
| Laboratory Testing | 24.5/28.0 | 88.0 % | 1 |
| Clinical Trials Oversight | 5.25/30.0 | 18.0 % | 1 |

**Fig 1. Quality of implementation and maturity level of medical device regulatory functions at the Medicines Control Authority of Zimbabwe between June and August 2022.** Regulatory functions were assessed to evaluate the strengths and weaknesses of MCAZ in regulating medical devices and IVDs. The quality of regulatory functions was implemented for condoms and gloves only. Other medical devices and IVDs have not been evaluated. The results are presented as proportions of implemented sub-indicators compared to the sum of the sub-indicators for each regulatory function. All regulatory functions were at maturity level 1.

the medicine framework. Subsequent amendments were implemented to include the formulation and implementation of regulation of condoms and gloves. The amendments did not cover all the regulatory functions for an effective regulatory system proportionate to the different health products. This is why the regulatory system for medical devices and IVDs is underdeveloped.

Currently, only condoms and gloves are regulated as medical devices, leaving other essential medical devices and IVDs unregulated. Hubner et al. further emphasize this inadequacy in regulation, highlighting the absence of defined regulations for medical devices in Zimbabwean legislation. Consequently, the MCAZ lacks a clear mandate to regulate these products. This leads to a deficiency in essential safety and performance standards and a lack of risk-based classification for devices, evidenced by attaining Maturity Level 1. This state of affairs aligns

with findings by Hubner et al. that Zimbabwe's regulatory system for medical devices and IVDs is not formal, indicating a lack of formal processes for regulation beyond condoms and gloves. The implications of these deficiencies are significant, as they potentially expose the Zimbabwean population to substandard and falsified medical devices and IVDs [6].

Comparatively, a study conducted by Dube-Mwedzi et al. underscores the ineffectiveness of the medical device regulatory system in Southern Africa compared to improvements observed in the regulatory frameworks for medicines. Moreover, the study suggests that the national regulatory systems of the member states in Southern Africa lack critical characteristics such as responsiveness, outcome orientation, predictability, risk proportionality to public health risk, and independence. The study presented findings that lacked granularity for each member state in the region. Therefore, it was challenging to compare Zimbabwe's state in that study directly with our current study. While the methodology used in our study differs from that of Dube-Mwedzi et al., which focused on a broader regional perspective, the findings remain consistent, emphasizing the urgency of addressing deficiencies in Zimbabwe's medical devices and IVDs regulatory system.

It is widely acknowledged that the effectiveness of medication is heavily reliant on accurate diagnosis, making the use of poor-quality medical devices and diagnostics detrimental to healthcare outcomes. Therefore, there is a pressing need for Southern African countries, including Zimbabwe, to prioritize strengthening regulatory frameworks for medical devices [11]. Using poor-quality medical devices and diagnostics undermines the effectiveness of effective medicines.

Looking at the Zimbabwean context, following an institutional approach that involves implementing the "best practices" seems plausible to harmonize regulations. The other option requires experimentation and prioritization of country-specific challenges, which may be challenging and costly for countries without experience. The problem-driven approach diverges from the institutional approach by prioritizing country-specific issues and enforcement over the blanket implementation of "best practices." This approach allows for feedback loops and greater policy experimentation as problems arise [12]. Therefore, the MCAZ must be empowered to regulate medical devices and IVDs with a solid legal foundation following the institutional approach.

Furthermore, this study examined registration and licensing (import and export controls), clinical testing, and post-market surveillance regulatory functions. In addition to the stated regulatory functions, the study conducted by this researcher looked further at indicators specific to regulatory systems, market authorization, licensing establishment, product performance evaluation, and regulatory inspection and quality management system audits.

Ultimately, following the institutional approach is crucial to empowering the MCAZ with a solid legal foundation to regulate medical devices and IVDs. Additionally, this study expands upon regulatory functions beyond registration and licensing, including clinical testing, post-market surveillance, market authorization, licensing establishment, product performance evaluation, and regulatory inspection, highlighting the need for comprehensive regulatory oversight in Zimbabwe.

The regulatory function of Registration and Market Authorization for medical devices and IVDs in Zimbabwe is currently below optimal performance due to the absence of legal provisions mandating registration before market access. Additionally, regulations and guidelines should support the notification or listing of low-risk medical devices and IVDs. The International Medical Devices Regulatory Framework proposes a classification system into four classes based on risk level, ranging from Class A (lowest risk) to Class D (highest risk), with corresponding levels of scrutiny for conformity assessment. However, the lack of evaluation means there is no assurance of the safety and effectiveness of medical devices and IVDs

currently in use [13–16]. Currently, there is no way to determine if medical devices and IVDs on the market and use are safe and effective, as they have not been evaluated.

Regarding Vigilance, the regulatory framework is also deficient, lacking legal provisions, regulations, and guidelines necessary to define and monitor the lifecycle of medical devices and IVDs. This absence puts the Zimbabwean population at risk of using products that do not meet safety, quality, and performance standards. A robust reporting system is essential to monitor such aspects and address risks such as substandard and falsified products and adverse effects from quality-assured medical devices and IVDs approved by other regulatory authorities. Examples from the United States Food and Drug Administration and the European Union highlight the importance of Vigilance, as adverse incidents have led to regulatory overhauls [17].

The discussion highlights several critical deficiencies in the regulatory framework for medical devices and IVDs in Zimbabwe, which pose significant risks to public health and economic stability. Firstly, the lack of a comprehensive system to monitor adverse events related to medical devices and IVDs implies that the true magnitude of these incidents remains unknown, potentially resulting in severe human and economic costs that go unnoticed and unmitigated. Legislation similar to that introduced by the European Union, which mandates economic operators to record, investigate, and report incidents and take Field Safety Corrective Actions, is necessary to ensure the safety and effectiveness of medical products in Zimbabwe [3, 4].

Market surveillance and control mechanisms are also inadequate. There are no legal provisions for regulatory interventions at entry and exit points or market surveillance activities specific to medical devices and IVDs. This exposes the population to substandard and falsified medical devices, compromising patient safety, the national economy, and public trust in the healthcare system. The lack of literature on this issue in Africa and the Middle East underscores the urgent need for action to ensure the quality and safety of medical devices and IVDs [18]. One study found that most countries in Southern Africa have legal authority to combat substandard and falsified medical products. Still, implementation is lacking, and challenges in managing medical products have been documented in the Southern African region [19]. The study was not specific to medical devices and IVDs but to medical products in general. It is unclear if Zimbabwe was amongst the countries with the legal authority, reiterating the need for more studies to be conducted in this area.

Furthermore, the absence of regulations for licensing economic operators along the supply chain of medical devices and IVDs means that products are not traceable in the value chain, except for condoms and gloves. Legislation should empower the MCAZ to issue licenses and oversee post-licensing changes, enhancing accountability and transparency in the supply chain [3, 4]. Licensing economic operators is pivotal in ensuring regulatory compliance and safeguarding the quality and safety of medical devices and IVDs. This licensing process encompasses various key aspects. Firstly, it ensures that all medical devices and IVD supply chain stakeholders, including manufacturers, authorized representatives, importers, and distributors, adhere to the regulatory standards stipulated in the legislation.

Additionally, licensing promotes traceability by assigning unique identifiers to economic operators, facilitating tracking of medical devices and IVDs from origin to end-users. This enhances transparency and accountability while bolstering regulatory oversight. Moreover, licensing entails the evaluation of economic operators' competence and capability to fulfil their obligations under the legislation, thereby upholding standards of quality assurance. Through this process, only qualified and reliable parties are authorized to distribute and market medical devices and IVDs, further bolstering consumer safety. Furthermore, licensing serves as a crucial mechanism for risk management by imposing regulatory obligations on economic operators, such as device conformity verification, incident reporting, and

documentation maintenance. This proactive approach enables the identification and mitigation of potential risks associated with IVD distribution more effectively. Lastly, licensing facilitates market surveillance by providing regulatory authorities with the means to monitor economic operators' activities, enforce compliance with the legislation, and investigate instances of non-compliance. This proactive regulatory framework ensures the protection of public health and safety while maintaining the integrity of the medical device and IVD market within the country.

The laboratory testing regulatory function for condoms and gloves was above average. This could be due to the ISO 17025:2017 accreditation and assessments by external organizations. Other medical devices and IVDs were not independently verified by laboratory testing based on the risk class of the medical device and IVD. Outsourcing laboratory testing is possible; however, laboratories' selection, monitoring, and evaluation should be clearly defined and documented. The evaluation can be performed through on-site audits using ISO 15189:2012 and ISO 17025:2017 requirements. An ISO accreditation is a good starting point, but more is needed, as anecdotal evidence shows that laboratories with ISO 15189:2012 accreditation failed the WHO audit for IVDs.

Performance evaluation is critical for verifying a manufacturer's claims regarding medical devices. Therefore, it is of paramount importance for the national regulatory authority to conduct an assessment and analysis of data to establish or verify the scientific validity, the analytical, and, where applicable, the clinical performance of a medical device to ensure that the intended purpose and classification are appropriate based on the specific disorder, condition, or risk factor of interest that it is intended to detect, define, or differentiate [3, 4]. Overall, performance evaluation is critical for ensuring the safety, effectiveness, and quality of medical devices and IVDs. It helps verify device claims, identify potential risks, comply with regulatory requirements, and drive continuous improvement in device performance, ultimately contributing to better patient outcomes and healthcare delivery. Currently, better patient outcomes and healthcare cannot be guaranteed due to a lack of verification of manufacturers' claims for medical devices and IVDs used in Zimbabwe by the MCAZ.

The MCAZ needs to have laboratories that it can use to conduct performance evaluations if it cannot do so. The laboratories must adhere to several essential measures to ensure accurate and reliable assessments. This includes employing personnel with expertise in relevant laboratory medicine, assay development, and data analysis. Establishing a robust quality management system specific to the scope of medical devices and IVDs is crucial, involving implementing standard operating procedures, quality control measures, and documentation protocols to maintain consistency and traceability. Laboratories should develop and adhere to validated protocols for validating and verifying medical devices and IVDs, ensuring accurate performance under intended conditions and populations. Adequate instrumentation and equipment are necessary for accurate evaluations, requiring investment in state-of-the-art technology and regular calibration. Utilization of appropriate reference materials and quality control samples is vital to validate and monitor test performance. Robust data analysis, interpretation, and reporting systems are essential, incorporating statistical methods and clinical context to prepare comprehensive reports for regulatory submission. Continual training and education of laboratory staff are paramount to maintain proficiency and stay updated on advancements in laboratory medicine and regulatory requirements. Implementing these measures ensures the accuracy, reliability, and integrity of performance evaluations conducted for regulatory approvals, enhancing the safety and effectiveness of medical devices and IVDs.

The current lack of legal provisions, regulations, and guidelines for regulatory inspections throughout the value chain of medical devices and IVDs presents a significant challenge. It is unclear whether other medical devices and IVDs establishments comply with applicable

standards, and the MCAZ must publish a list of measures that all economic operators must follow to ensure compliance. Firstly, Inspections ensure compliance by verifying adherence to requirements among manufacturers, authorized representatives, importers, and distributors. This includes assessing compliance with quality management systems, conformity assessment procedures, performance evaluation requirements, and post-market surveillance obligations, thus enabling corrective actions to rectify deviations or non-compliance issues.

Secondly, inspections assess the quality and safety of medical devices and IVDS by evaluating manufacturing processes, quality control measures, and product documentation. This scrutiny ensures that IVDs are manufactured, distributed, and marketed following established quality standards and regulatory specifications, minimizing the risk of harm to patients and users.

Thirdly, inspections verify the performance and clinical validity of IVDs through assessments of performance evaluation studies and clinical evidence supporting device claims. This scrutiny ensures that IVDs provide accurate and reliable diagnostic information, contributing to improved patient outcomes.

Moreover, regulatory inspections promote transparency and accountability within the medical devices and IVD industries by providing independent oversight and validation of manufacturers' activities. Regular inspections and publication of inspection reports demonstrate regulatory commitment to upholding standards and protecting public health, fostering trust among stakeholders.

Lastly, inspections are a proactive mechanism for detecting and addressing non-compliance issues before they escalate. By identifying deficiencies early on, regulatory authorities can collaborate with manufacturers to implement corrective and preventive measures, mitigating potential risks and improving overall compliance. Overall, regulatory inspection plays a vital role in ensuring the quality, safety, and effectiveness of IVDs, ultimately benefiting public health and safety.

The oversight of clinical trials for medical devices and IVDs is not covered by Zimbabwe's Medicines and Allied Substances Control Act. No legal framework mandates the MCAZ to review clinical trial protocols for medical devices and IVDs. To ensure the safety and ethical conduct of clinical trials, it is recommended that trained staff conduct a protocol review before the trial begins, and an Independent Ethics Committee should review and revise the protocol if necessary. The Clinical Trial Review Committee should comprise competent members without conflicts of interest. Currently, the MCAZ lacks the capacity to review clinical trial protocols, inspect clinical trial sites, or take action against products that do not meet the required standards. Moreover, the MCAZ may not be aware of medical devices and IVDs that are undergoing clinical trials. This poses a significant risk to the rights of participants in these trials, as they may not be informed about the devices' registration status, quality, safety, and performance. The history of violations of patient rights in research, such as the Tuskegee Syphilis Study, Nazi medical experimentation, and research at Willowbrook State School, underscores the importance of regulations governing the design and implementation of human-subject research protocols [20].

To effectively regulate medical devices and IVDs in Zimbabwe, The Medicines and Allied Substances Control Act needs to be amended to incorporate key components into the legislative framework:

1. **Definition and Classification:** Clearly define medical devices and IVDs, establishing a risk-based classification system for regulatory oversight.

2. **Regulatory Authority Establishment:** Empower the MCAZ with adequate resources and expertise to enforce regulatory requirements effectively.

3. **Licensing and Registration:** Outline requirements for the licensing or registration of manufacturers, importers, and distributors, ensuring compliance with quality management systems and post-market surveillance obligations.

4. **Quality Management Systems:** Mandate implementation of quality management systems by manufacturers, adhering to specified standards and guidelines, with regular audits and inspections for compliance verification.

5. **Conformity Assessment:** Establish procedures for conformity assessment to ensure devices meet regulatory requirements, including pre-market approval, product testing, and clinical evaluations.

6. **Post-Market Surveillance:** Mandate post-market surveillance systems for monitoring device safety and performance, with provisions for adverse event reporting, recalls, and periodic re-evaluations.

7. **Labeling and Instructions for Use:** Require clear labeling and instructions for use to ensure safe and effective device utilization by healthcare professionals and end-users.

8. **Capacity Building and Training**: Include provisions for capacity building and training initiatives to enhance regulatory capacity among stakeholders, including regulatory authorities, healthcare professionals, and industry representatives. The competence framework must be put in place that includes proficiency in scientific and health concepts, encompassing the understanding and applying evolving basic and translational science, regulatory science, and public health principles to enhance healthcare product development, evaluation, and oversight [21–23].

9. **International Harmonization:** Align legislative provisions with international standards and guidelines to facilitate trade and harmonization efforts, potentially adopting regulations developed by the International Medical Devices Regulators Forum and WHO.

Incorporating these components into the legislative framework enables Zimbabwe to establish a robust regulatory system for medical devices and IVDs, ensuring their safety, quality, and effectiveness in healthcare delivery.

## Study limitations

The present study was performed with the purview of MCAZ as the national regulatory authority. However, the Medical Laboratory and Clinical Scientists Council of Zimbabwe and the National Microbiology Reference Laboratory were not included in the examination. These two institutions are vital in registering IVDs for the priority pathogens. The registration of IVD medical devices was used as a criterion for products to be procured through the national tendering process. However, these two institutions do not have a legal mandate to be national regulatory authorities.

## Conclusions

The existing legal provisions, regulations, and guidelines governing medical device and IVD regulations are deemed insufficient, as the review highlights deficiencies in the regulatory system's robustness, suitability for its intended purpose, responsiveness, transparency, and alignment with the risk classification of medical devices and IVD. It is crucial to amend the Medicines and Allied Substance Control Act to incorporate the definition and classification of medical devices and IVDs, regulatory authority establishment, licensing and registration, quality management system, conformity assessment, post-market surveillance, labeling and

instructions for use, capacity building and training, and international harmonization. This is the first step in ensuring a robust regulatory system for medical devices and IVDs.

## Supporting information

**S1 Table. List of documents reviewed.**
(TIF)

## Acknowledgments

The authors acknowledge the cooperation of the Medicines Control Authority of Zimbabwe.

## Author Contributions

**Conceptualization:** Charles Chiku.

**Data curation:** Charles Chiku.

**Formal analysis:** Charles Chiku.

**Investigation:** Charles Chiku.

**Methodology:** Charles Chiku.

**Project administration:** Charles Chiku.

**Resources:** Charles Chiku.

**Supervision:** Talkmore Maruta, Justen Manasa.

**Validation:** Charles Chiku.

**Visualization:** Charles Chiku.

**Writing – original draft:** Charles Chiku.

**Writing – review & editing:** Charles Chiku, Talkmore Maruta, Fredrick Mbiba, Justen Manasa.

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
