## [Decision Letter · Decision Letter 0]

20 Oct 2023

PONE-D-23-17232Medical Devises Regulation in Zimbabwe: An Evaluation of operational readinessPLOS ONE

Dear Dr. Chiku,

Thank you for submitting your manuscript to PLOS ONE. After careful consideration, we feel that it has merit but does not fully meet PLOS ONE’s publication criteria as it currently stands. Therefore, we invite you to submit a revised version of the manuscript that addresses the points raised during the review process. In this process, pay particular attention to the need to carry out a professional review of English vocabulary and grammar, as well as to provide a more detailed description of the methodology.

We look forward to receiving your revised manuscript.

Kind regards,

Zenewton André da Silva Gama, Ph.D.

Academic Editor

PLOS ONE

Reviewers' comments:

Reviewer's Responses to Questions

**Comments to the Author**

1. Is the manuscript technically sound, and do the data support the conclusions?

Reviewer #1: Yes

Reviewer #2: Partly

2. Has the statistical analysis been performed appropriately and rigorously? 

Reviewer #1: N/A

Reviewer #2: N/A

3. Have the authors made all data underlying the findings in their manuscript fully available?

Reviewer #1: Yes

Reviewer #2: No

4. Is the manuscript presented in an intelligible fashion and written in standard English?

Reviewer #1: Yes

Reviewer #2: No

5. Review Comments to the Author

Reviewer #1: Summary of the research and overall impression:

The study reports an analysis of the readiness of medical device regulation in Zimbabwe according to the Global Benchmarking Tool+ provided by World Health Organization. The manuscript highlights how regulation of medical devices has seriously lagged not only in Zimbabwe but in low-income countries compared to EU regulation or FDA in US. Results report that, according to the proposed framework, maturity level of medical devices regulation in Zimbabwe is 1, underlining the lack of guidelines by regulatory agencies in all the phases of the management cycle of medical devices including post-market surveillance and control. In the opinion of this reviewer the article topic is really relevant. Technical aspects of the study design and data analysis are well detailed and documented. However, some parts need to be rearranged in a more accessible way.

The authors should address the following issues:

Major Revisions

1. Please correct the misspelling in the title about the “devises” word.

2. Please review the English grammar and vocabulary in the entire manuscript.

3. Some concepts are repeated a lot of times, with same words and expressions. Please consider eliminating repetitions especially in the results discussion section.

4. An extensive number of acronyms are used in the text, although they are necessary a large number of acronyms make the manuscript difficult to read. It is advisable to avoid them where possible. Some acronyms are not necessary because they are not frequently recalled in the text.

Minor Revisions:

5. Image captions are very poor in content. Please include more information about reported graphs in figure 1 and figure 2.

6. Line 10-11: Please rephrase “There are disparities in regulating medical devices; in the African region, it is below the global average”.

7. Line 67: Developing countries is a term disused. Please substitute with low-income countries or any term you consider appropriate to this context.

8. Line 93-94: “About 40% of countries in the WHO Afro region had no regulations for medical devices, 32% had some regulations, and 28% had no regulations”. Please rephrase, it is not clear how percentages are distributed.

9. Line 123: Include in the manuscript NRAs acronym definition. It is present only in the abstract.

10. Line 301-302: “No requirements require medical devices..” Please rephrase.

11. Line 350: “RS” please avoid the use of acronyms also in the titles.

12. Line 378: Please instead of cite the single author (Dube-Mwedzi) cite the published journal article.

13. Line 401: “MA” please avoid the use of acronyms also in the titles.

14. Line 406: Please consider that GHTF is the acronym referring to global harmonization task force, not the global harmonization working party.

15. Line 418: “VL” please avoid the use of acronyms also in the titles.

16. Line 439: “MC” please avoid the use of acronyms also in the titles.

17. Line 461-462: “Still they were not doing so” please rephrase.

18. Line 462: US acronym is already used. Please redefine at the beginning of the manuscript. Same suggestion can be addressed for European Union.

19. Line 468: “LI” please avoid the use of acronyms also in the titles.

20. Line 478: “LT” please avoid the use of acronyms also in the titles.

21. Line 480: Please rephrase clarifying what it is included what it is not.

22. Line 492: Please also insert the year in the ISO standards.

23. Line 495: Please also insert the year in the ISO standard.

24. Line 503: “RI” please avoid the use of acronyms also in the titles.

25. Line 514: “CT” please avoid the use of acronyms also in the titles.

26. When regulation are cited, it is better to refer directly to the regulation instead of a document reporting the regulation.

27. Citations 14 and 15 are the same paper.

28. Please check that the citation style is uniform.

Reviewer #2: This is an important paper about the status of medical device regulation in Zimbabwe. The paper requires several revisions.

1. The introduction should be made more concise.

2. The methods are unclear. It says that a qualitative study was performed using WHO GBT+ methodology. What does this mean? There is no reference in the methods to a paper or website describing this methodology. Furthermore, the paper should clearly describe the methods without the reader needing to read another paper to understand what was done. It says that regulatory oversight was evaluated from June - August 2022, but what does this mean? Did someone track all medical devices during that time? Did the authors interview people at the NRA? Did they review documents? Did they review policies? How were the different regulatory functions assessed? How many medical devices were included in this study?

It is difficult to review this paper, when the reader cannot completely understand the methods and what was done in the study.

3. Figure 1 does not make sense to me. How did you determine what was estimated versus what was implemented? And if this is real data, it should not appear so perfectly symmetrical / round. Shouldn't there be some variation in different parameters performing better or worse?

4. Figure 2 I also do not understand what was measured here, and how the authors came up with this data.

Overall, the paper needs significant work and restructuring so that the methods and results are clearly stated. Then a proper review should be repeated.

6. PLOS authors have the option to publish the peer review history of their article (what does this mean?). If published, this will include your full peer review and any attached files.

Reviewer #1: No

Reviewer #2: No

---

## [Author Response · Author response to Decision Letter 0]

8 Nov 2023

Response to Reviewers’ comments for the study PONE-D-23-17232

Medical Devises Regulation in Zimbabwe: An Evaluation of operational readiness.

Thank you very much for your time and effort in reviewing the manuscript with the title on the subject line. I am grateful for the invaluable feedback you and the other reviewers provided to improve my work. I responded to each question raised during the review process. Where I needed to explain the rationale for the approach I used, I have done so as well. Each question had a corresponding response. The questions from the reviewers are indented and in bold to distinguish them from the responses provided.

General Comments in Email

Response: The manuscript was edited to meet PLOS ONE’S style requirements, including file naming requirements. 

Response: Data was deposited in Open Science Framework (OSF) Registries under the Project name Medical devices regulation in Zimbabwe: an evaluation of operational readiness, doi: https://doi.org/10.17605/OSF.IO/C5JXG.

3. Please include captions for your Supporting Information files at the end of your manuscript and update any in-text citations to match accordingly. Please see our Supporting Information guidelines for more information: http://journals.plos.org/plosone/s/supporting-information.

Response: There are no supporting information files attached to the manuscript. The files added in the initial submission as Supporting files 1 and Supporting file 2 were raw data. These files were deposited in the OSF Registries at https://doi.org/10.17605/OSF.IO/C5JXG.

Major Revisions 

Reviewer 1

Reviewers’ comment 

1. Please correct the misspelling in the title about the “devises” word.

Response: The spelling is corrected in the title and throughout the manuscript. Please see the title in both the clean and tracked manuscript versions. 

Reviewers’ comment

2. Please review the English grammar and vocabulary in the entire manuscript.

Response: The entire manuscript was reviewed and corrected to address all the identified English grammar and vocabulary errors identified by the reviewers. The tracked changes version of the manuscript shows the corrections that addressed the grammatical and vocabulary errors. 

Reviewers’ comment

3. Some concepts are repeated a lot of times, with the same words and expressions. Please consider eliminating repetitions, especially in the results and discussion section.

Response: Repeated concepts, such as registration and market authorization, dealing with substandard and falsified products, were revised, and repetitions were eliminated. 

Reviewers’ comment

4. An extensive number of acronyms are used in the text; although they are necessary, a large number of acronyms make the manuscript difficult to read. It is advisable to avoid them where possible. Some acronyms are not necessary because they are not frequently recalled in the text.

Response: Acronyms were reduced and referred to regulatory functions in full instead of using acronyms. Please refer to lines 454-572.

Minor Revisions:

Reviewers’ comment

5. Image captions are very poor in content. Please include more information about reported graphs in figure 1 and figure 2.

Response: Figures 1 and 2 were combined into a single figure, named Fig 1, in the revised submission. Revisions include the legend and caption to give the readers a better understanding of the results. The legend was added in the manuscript just below the title of the figure to meet the requirements of PLOS One journal as stated in the submission guidelines. 

Reviewers’ comment: 

6. Line 10-11: Please rephrase“There are disparities in regulating medical devices; in the African region, it is below the global average”.

Response: The statement was corrected and read as, “There are disparities in the regulation of medical devices in the African region that are below the global average.” (Line 33). 

Reviewers’ comment: 

7. Line 67: Developing countries is a term disused. Please substitute with low--income countries or any term you consider appropriate to this context.

Response: The word developing countries was replaced with RLS (Resource Limited Settings). Lines 90-95, “ It has also been concluded that poor regulatory systems in RLS make it difficult for manufacturers to introduce their products into these markets, resulting in limited access to them.”

Reviewers’ comment

8. Line 93-94: “About 40% of countries in the WHO Afro region had no regulations for medical devices, 32% had some regulations, and 28% had no regulations”. Please rephrase, it is not clear how percentages are distributed.

Response: This section was reviewed to report the source of information in the context of the desk review conducted by the World Health Organization. Criteria on the classification of regulatory systems in the surveyed member states were also included, as shown below: “ The WHO conducted a desk review in 2016 to assess the availability of medical device legislation in Africa. Countries were classified based on premarket, market, and post-market regulatory elements as follows:

• Have all the three types of elements,

• Have premarket and placing on the market elements 

• Have premarket and post-market elements,

• Have placing on the market and post-market elements

• Have only premarket elements 

• Have only placing on the market elements 

• Do not have any element

• Data not available (member states did not respond to the survey). 

• Not applicable 

In the Afro region, 40% of countries have no medical device regulations, 32% have some regulations, and 28% lack data. Globally, 58% of the WHO members regulate these devices. Zimbabwe was reported as regulating medical device market placement, but data is vague (12) .” (lines 115-130).

Reviewers’ comment

9. Line 123: Include in the manuscript NRAs acronym definition. It is present only in the abstract.

Response: The acronym NRA was mentioned in full at the first mention in the body, as shown in lines 202-204, “Manufacturers may be hesitant to introduce their products in countries with weak regulations if the National Regulatory Authority (NRA) has additional requirements compared to mature regulatory authorities, where these products may have been authorized.”

Reviewers’ comment

10. Line 301-302:“No requirements require medical devices..” Please rephrase.

Response: The statement was corrected: “Medical devices lack legal compliance for registration and authorization. No conformity assessments were proportional to the risk class of medical devices. " (Lines 366-367). 

Reviewers comment: 

11. Line 350: “RS” please avoid the use of acronyms also in the titles.

Response: Acronyms were removed from all the titles in the results and discussion section of the manuscript as follows: Regulatory System (line 395), Registration and Market Authorisation (line 454), Vigilance (line 477), Market Surveillance and Control (line 507), Licensing Establishment (532), Laboratory Testing (line 543), Regulatory Inspection (line 561), and Clinical Trials Oversight (572). 

Reviewers’ comment

12. Line 378: Please, instead of cite the single author (Dube-Mwedzi), cite the published journal article.

Response: The journal article was cited as “This agrees with a study conducted by Dube-Mwedzi et al., which found that the regulatory frameworks for medicines were strengthened, while medical devices were not a priority in the rapid assessment of the National Regulatory Systems for medical products in the Southern African Development Community, which clearly shows that Zimbabwe's medical device regulatory system has not been responsive, outcome-oriented, predictable, risk proportionate to public health risk, and independent; these are critical characteristics of a robust regulatory system. This finding is in agreement with that of Dube-Mwedzi et al. (12). Poor quality medical devices have a significant impact on health care outcomes. It is widely accepted that the value of medication relies mainly on the accuracy of diagnosis. Thus, the use of poor-quality medical devices and diagnostics undermines the effectiveness of effective medicines. Investing more effort in strengthening frameworks for medical devices is recommended as a priority for SADC countries. 

The methodology used in the rapid assessment mentioned above differed from this study in that Dube-Mwedzi et al.'s focused on the SADC member states, and the data lacked granularity to identify where Zimbabwe stood in the study. In our study, we conducted an in-depth examination of the Zimbabwean context using a global benchmarking tool to assess the quality of implementation of the medical device regulatory system.”(lines 418-435). 

Reviewers’ Comment: 

13. Line 401: “MA” please avoid the use of acronyms also in the titles.

Response: The acronym was replaced with Registration and Market Authorisation (line 454). 

Reviewers’ comment: 

14. Line 406: Please consider that GHTF is the acronym referring to global harmonization task force, not the global harmonization working party.

Response: My apologies for the typographical error. The statement has been corrected as “The International Medical Devices Regulatory Framework (IMDRF) and its predecessor, the Global Harmonization Task Force (GHTF), recommend that medical devices be assigned to one of four classes based on a set of rules..” (lines 467-449). 

Reviewers comment: 

15. Line 418: “VL” please avoid the use of acronyms also in the titles.

16. Line 439: “MC” please avoid the use of acronyms also in the titles.

Response: Please refer to the response to request 11. 

Reviewers’ comment

17. Line 461-462: “Still they were not doing so” please rephrase.

Response: The statement was rephrased: "One study found that most countries in the region have legal authority to combat substandard and falsified medical products, but implementation is lacking, and challenges in managing medical products have been documented in the SADC region. However, more information is required regarding medical devices, including IVDs. A study conducted to map existing frameworks, mechanisms and approaches to prevention, detection and response to substandard and falsified medical products concluded that most countries had the legal mandate to implement measures to curb substandard and falsified medical products. Still, they were not implemented (13).” (lines 523-530). 

Reviewers comment: 

18. Line 462: The US acronym is already used. Please redefine at the beginning of the manuscript. Same suggestion can be addressed for European Union.

Response: The acronym was redefined for both the United States and European Union in lines 86-88 as shown as “Regulatory processes in the African region are not as well documented as those in the United States Food and Drug Administration (USFDA) and European Union (EU) European Medicines Agency (5).”

Reviewers comment: 

19. Line 468: “LI” please avoid the use of acronyms also in the titles.

20. Line 478: “LT” please avoid the use of acronyms also in the titles.

Response: Please refer to the response to request 11 above. 

Reviewers’ comment:

21. Line 480: Please rephrase clarifying what it is included what it is not.

Response: The statement was rephrased to show that only gloves and condoms were tested to verify their safety and performance before marketing authorization. Other medical devices and IVD medical devices were not subjected to testing as part of the conformity assessment to be granted market authorization. The statement was rephrased: " The laboratory testing regulatory function scored high, with 88% for condom and glove testing, as the laboratory was ISO 17025 accredited. However, this score may be due to external body assessments, leading to an established and effective quality management system. Medical devices other than condoms and gloves were not independently verified by laboratory testing based on the risk class of the medical device.” (lines 544-548). 

Reviewers’ comments

22. Line 492: Please also insert the year in the ISO standards.

Response: The year the ISO standards were issued were included for ISO 15189:2012 and ISO 17025:2017. The statement reads, “The evaluation can be performed through on-site audits using ISO 15189:2012 and ISO 17025:2017 requirements. Having an ISO accreditation is a good starting point, but more is needed, as anecdotal evidence shows that laboratories with ISO 15189:2012 accreditation failed the WHO audit for IVDs.” (lines 550-553). 

Reviewers comment

23. Line 495: Please also insert the year in the ISO standard.

Response: The year of the ISO 15189 standard was added and read as follows, “The evaluation can be performed through on-site audits using ISO 15189:2012 and ISO 17025:2017 requirements. Having an ISO accreditation is a good starting point, but more is needed, as anecdotal evidence shows that laboratories with ISO 15189:2012 accreditation failed the WHO audit for IVDs.” (lines 550-553). 

Reviewers’ comment

24. Line 503: “RI” please avoid the use of acronyms also in the titles.

Response: Please refer to the response to request 11 above.

Reviewers’ comment

25. Line 514: “CT” please avoid the use of acronyms also in the titles.

Response: Please refer to the response to request 11 above.

Reviewers’ comment:

26. When regulations are cited, it is better to refer directly to the regulation instead of a document reporting the regulation.

Response: It is unclear where a document reporting the regulation was cited instead of the source. Regulations that were cited were the European Union Medical Devices Regulation (Regulation (EU) 2017/745 of The European Parliament and of the Council on Medical Devices) and In Vitro Diagnostic Medical Devices Regulations (IVDR- Regulation (EU) 2017/746). These regulations are published in the European Journal, the primary source used for citing (European Commission. 

3. Regulation (EU) 2017/745 of The European Parliament and of 620 the Council on Medical Devices. Off J Eur Union. 2017;5(8):175. 621

4. European Commission. Regulation (EU) 2017/746 of the European Parliament and of 622 the Council on in vitro diagnostic medical devices. Off J Eur Union. 2017;5(5):117–76.)

 Could you kindly indicate where a regulation was cited from a secondary source so I can address it accordingly? 

Reviewers’ comment: 

27. Citations 14 and 15 are the same paper.

Response: The citation was corrected. The duplicated citation was deleted and replaced with the correct one. 

Reviewers; comment

28. Please check that the citation style is uniform.

Response: Citations were checked accordingly and corrected using the Vancouver style. 

Reviewer 2:

Reviewer #2: This is an important paper about the status of medical device regulation in Zimbabwe. The paper requires several revisions.

Reviewer Comment: 

1. The introduction should be made more concise.

Response: The introduction was reviewed to make it more concise. Please refer to the introduction in the manuscript (lines 69-221). 

Reviewer comment: 

2. The methods are unclear. It says that a qualitative study was performed using WHO GBT+ methodology. What does this mean? There is no reference in the methods to a 

---

## [Decision Letter · Decision Letter 1]

14 Dec 2023

PONE-D-23-17232R1Medical devises regulation in Zimbabwe: An evaluation of operational readiness.PLOS ONE

Dear Dr. Chiku,,

Thank you for submitting your manuscript to PLOS ONE. After careful consideration, we feel that it has merit but does not fully meet PLOS ONE’s publication criteria as it currently stands. Therefore, we invite you to submit a revised version of the manuscript that addresses the points raised during the review process.

The article has had important improvements, but needs to follow the reviewers' latest recommendations for it to be accepted.

We look forward to receiving your revised manuscript.

Kind regards,

Zenewton André da Silva Gama, Ph.D.

Academic Editor

PLOS ONE

Reviewers' comments:

Reviewer's Responses to Questions

**Comments to the Author**

1. If the authors have adequately addressed your comments raised in a previous round of review and you feel that this manuscript is now acceptable for publication, you may indicate that here to bypass the “Comments to the Author” section, enter your conflict of interest statement in the “Confidential to Editor” section, and submit your "Accept" recommendation.

Reviewer #1: (No Response)

Reviewer #2: (No Response)

2. Is the manuscript technically sound, and do the data support the conclusions?

Reviewer #1: Yes

Reviewer #2: Partly

3. Has the statistical analysis been performed appropriately and rigorously? 

Reviewer #1: N/A

Reviewer #2: N/A

4. Have the authors made all data underlying the findings in their manuscript fully available?

Reviewer #1: Yes

Reviewer #2: Yes

5. Is the manuscript presented in an intelligible fashion and written in standard English?

Reviewer #1: Yes

Reviewer #2: No

6. Review Comments to the Author

Reviewer #1: The reviewer thanks the authors for addressing the revisions suggested. However, the authors should address the following issues:

Major revisions:

1. Line 275: “Two participants were interviewed and assisted with information to complete the questionnaires administered by the researcher.” Were only two participants interviewed? Which are the inclusion criteria on which they were chosen?

2. Introduction section is still long. The reviewer’s suggestion is to maintain only the information necessary to have an overview of the problem addressed by the manuscript.

3. Grammar revision has been not effective. Here some examples:

“Medical devices are essential for the public to meet their highest health standards” instead of “Medical devices are essential for the public to reach the highest health standards.”

“Medical Device Regulations are a set of laws and regulations governing the clinical trials,

manufacturing, and distribution of medical devices to ensure that they are safe and perform as intended by their manufacturers” Instead of “Medical Device Regulations are a set of laws and regulations governing clinical trials, manufacturing, and distribution of medical devices ensure they are safe and perform as intended by their manufacturers”.

“Participation in the study was voluntary” instead of “Participation was voluntary”

“The participants were able to stop the interview at any time without any explanation” instead of “Participants were able to stop the interview at any time without explanation”

These examples are not exhaustive.

Minor revisions:

4. Image 1 is low quality. In addition, in the opinion of the reviewer the image does not add any value to the manuscript.

5. Line 117-118: “In 2016, the World Health Organization (WHO) conducted a 117 review to evaluate the availability of medical device regulations in Africa.” Are you sure only Africa was the target area of the study? Probably you are referring to WHO regulatory status desk survey (July 2016 update) and this study was targeted to map all the world status. Please even if this is not the study you have analyzed, cite in the text which document you are referring to.

6. Line 146: ” countries of the Surgeons of Central, Eastern and Southern Africa (COSECA)”, the acronym was already defined.

7. Line 228: “The researcher interviewed the staff responsible for the regulations and conducted a desk reviewed the documents.” Please check the grammar or rephrase.

8. Line 548: Please insert the year of the standard.

9. The reviewer suggests to cut all the parts of the manuscript that really do not refer to the main scope and to insert them as supplementary material.

10. Answer to comment 26 of the first review. Reference n.2 (GHTF. Global Harmonization Task Force Study Group 1: Definition of the Terms ‘Medical Device’ and ‘In Vitro Diagnostic (IVD) Medical Device.’ Force, Study Gr 1Glob Harmon Task [Internet]. 2012;(Ivd):6. Available from:http://www.imdrf.org/docs/ghtf/final/sg1/technical-docs/ghtf-sg1-n071-2012 definition-of-terms-120516.pdf#search=%22ghtf definition ?Medical Device?

is not necessary as you can refer directly to MDR 745/2017 and IVDR 746/2017 for the definitions of medical device and in-vitro diagnostic medical device.

Line numbers refer to the version without track changes.

Reviewer #2: The authors have made several changes that have improved the paper. However, more changes are needed.

1. The authors refer to Resource Limited Settings (RLS). Please just use the term "LMIC" - low- and middle-income countries, as the other reviewer suggested, as this is the appropriate term. Most countries in Africa are not without resources. There are actually a lot of resources (diamonds, coffee, oil, etc...). The use and corruption of these resources is complicated. So please just stick with LMICs.

2. The other reviewer has complained about too many abbreviations, and there are still too many abbreviations / acronyms. I would limit the paper to 3-5 acronyms and then spell out everything else. The reader cannot keep track of all of the acronyms.

3. The introduction is still significantly too long. The authors need to express the same content, but use half as many words. There is a lot of redundant information in the introduction.

4. In general, the paper is about a very important topic and should be published. However, the authors need significant help from an experienced author to restructure the paper to make it more readable. In its current form, it is too repetitive in some areas, and then does not give adequate details in the methods and results sections. I would encourage the authors to seek out a global health partner, who has more experience publishing scientific manuscripts to help them with the structure of the paper.

7. PLOS authors have the option to publish the peer review history of their article (what does this mean?). If published, this will include your full peer review and any attached files.

Reviewer #1: No

Reviewer #2: No

---

## [Author Response · Author response to Decision Letter 1]

28 Jan 2024

Major revisions:

Reviewers’ Comment: 

1. Line 275: “Two participants were interviewed and assisted with information to complete the questionnaires administered by the researcher.” Were only two participants interviewed? Which are the inclusion criteria for which they were chosen? 

Response: 

Thank you for the comment. Indeed, the inclusion criteria for selecting participants was not defined in the previous version of the manuscript. The sampling method and inclusion criteria were amended as follows, “The inclusion criteria comprised three main categories: Firstly, officials directly engaged in regulatory decision-making were considered integral to providing insights into the decision-making processes. Secondly, individuals with substantial expertise and experience in health product regulation were targeted, ensuring a wealth of knowledge and a nuanced understanding of the regulatory landscape. Lastly, individuals responsible for critical regulatory functions, such as product registration, licensing, quality control, inspections, pharmacovigilance, and post-marketing surveillance, were included to capture diverse perspectives and experiences within the regulatory domain. Through this careful selection process, the study aimed to gather a well-rounded and insightful perspective on health product regulation from those actively shaping and implementing regulatory policies.” (lines 155-165).

Reviewer’s comment

2. Introduction section is still long. The reviewer’s suggestion is to maintain only the information necessary to have an overview of the problem addressed by the manuscript.

Response: The introduction was reduced by removing some of the content not specific to the Zimbabwean context. (See lines 71-126). 

Reviewer’s comment

3. Grammar revision has been not effective. Here some examples:

“Medical devices are essential for the public to meet their highest health standards”instead of “Medical devices are essential for the public to reach the highest health standards.”

“Medical Device Regulations are a set of laws and regulations governing the clinical trials,

manufacturing, and distribution of medical devices to ensure that they are safe and perform asintended by their manufacturers”Instead of “Medical Device Regulations are a set of laws and regulations governing clinical trials, manufacturing, and distribution of medical devices ensure they are safe and perform as intended by their manufacturers”.

“Participation in the study was voluntary” instead of “Participation was voluntary”

“The participants were able to stop the interview at any time without any explanation ”instead of “Participants were able to stop the interview at any time without explanation”

These examples are not exhaustive.

Response: Thank you for the comments. The examples were addressed accordingly; see lines 71-72, “Medical devices and In Vitro Diagnostic (IVD) medical devices are essential for the public to reach their highest health standards.”

 Lines 86-88, “Medical device and IVD regulations are laws that govern the design and development, manufacturing, clinical trials, and distribution of medical devices. The aim is to ensure that they are safe and perform as intended by their manufacturers (3,4).”

Lines 220-221: “Participation in the study was voluntary. The participants were able to stop the interview at any time without any explanation, if necessary.” 

Furthermore, the manuscript was reviewed extensively to correct the grammatical and typographical errors. 

Minor revisions:

Reviewer’s comment 

4. Image 1 is low quality. In addition, in the opinion of the reviewer the image does not add any value to the manuscript.

Response: The image may be of low quality, but it is still valuable as it provides the readers with an overview of the quality of implementation and the maturity level of each regulatory function that applies to medical devices and IVDs' regulatory system. This is the key output of benchmarking conducted by regulatory authorities. Therefore, removing it, as suggested by the reviewer, defeats the whole purpose of conducting the benchmarking. Regarding the quality of the figure, we used the pace application to create an image as required by Plos One journal. Therefore, the authors have decided to keep the image as part of the manuscript. 

Reviewer’s comment

5. Line 117-118:“In 2016, the World Health Organization (WHO) conducted a 117 review to evaluate the availability of medical device regulations in Africa.” Are you sure only Africa was the target area of the study? Probably you are referring to WHO regulatory status desk survey (July 2016 update) and this study was targeted to map all the world status. Please even if this is not the study you have analyzed, in the text which document you are referring to.

Response: It was a typographical error. The referenced source (https://www.who.int/teams/health-product-policy-and-standards/assistive-and-medical-technology/medical-devices/regulations) referred to the global Desk Review of Medical Device Regulatory Systems. The statement was rephrased to read as follows, “In 2016, the World Health Organization (WHO) conducted a review to evaluate the availability of medical device regulations globally.” (line 93). 

Reviewer’s comment: 

6. Line 146:” countries of the Surgeons of Central, Eastern and Southern Africa (COSECA)”, the acronym was already defined.

Response: We amended to state the abbreviation after first being mentioned in line 106 (the amendment is in line 120 and 124). 

Reviewers’ comment

7. Line 228: “The researcher interviewed the staff responsible for the regulations and conducted a desk reviewed the documents.” Please check the grammar or rephrase.

Response: The statement was rephrased: "A total of two participants were interviewed. Additionally, an extensive desk review of pertinent documents was conducted to validate the current status of legal provisions, regulations, and guidelines governing the regulatory landscape. This comprehensive approach ensured a nuanced understanding of the regulatory framework within the MCAZ.”(lines 165-169).

Reviewer’s comment

8. Line 548: Please insert the year of the standard.

Response: “The laboratory testing regulatory function for condoms and gloves was above average. This could be due to the ISO 17025:2017 accreditation and assessments by external organizations. Other medical devices and IVDs were not independently verified by laboratory testing based on the risk class of the medical device. Outsourcing laboratory testing is possible; however, laboratories' selection, monitoring, and evaluation should be clearly defined and documented. The evaluation can be performed through on-site audits using ISO 15189:2012 and ISO 17025:2017 requirements.” (lines 420 and 425). 

Reviewer’s comment

9. The reviewer suggests to cut all the parts of the manuscript that really do not refer to the main scope and to insert them as supplementary material.

Response: The introduction was reduced by removing the context of the regulatory system in the United States of America and the European Union, Convergence, Reliance and Recognition Mechanisms and those for the Southern African Development Community region without granular data for Zimbabwe. 

Reviewer’s comment

10. Answer to comment26 of the first review. Reference n.2(GHTF. Global Harmonization Task Force Study Group 1: Definition of the Terms ‘Medical Device’ and ‘In Vitro Diagnostic (IVD) Medical Device.’ Force, Study Gr 1Glob Harmon Task [Internet]. 2012;(Ivd):6. Available from:http://www.imdrf.org/docs/ghtf/final/sg1/technical-docs/ghtf-sg1-n071-2012definition-of-terms-120516.pdf#search=%22ghtf definition ?Medical Device?

is not necessary as you can refer directly to MDR 745/2017 and IVDR746/2017 for the definitions of medical device and in-vitro diagnostic medical device.

Line numbers refer to the version without track changes.

Response: Thank you for pointing out where regulations should have been cited directly. However, the Global Harmonization Task Force is the primary source that defines Medical Devices and In Vitro Diagnostic Medical Devices from which regulatory authorities such as the European Union’s MDR 745/2017 and IVDR 746/2017 adopted their definition. The MDR 745/2017 and IVDR746/2017 regulations were published in 2017, which were later compared to the GHTF guidance document published in 2012. In this regard, the authors will maintain this citation and reference as in the previous version of the manuscript. 

Reviewer’s comment 

Reviewer #2: The authors have made several changes that have improved the paper. However, more changes are needed.

1. The authors refer to Resource Limited Settings (RLS). Please just use the term "LMIC" - low- and middle-income countries, as the other reviewer suggested, as this is the appropriate term. Most countries in Africa are not without resources. There are actually a lot of resources (diamonds, coffee, oil, etc...). The use and corruption of these resources is complicated. So please just stick with LMICs.

Response: Thank you for the comment and for explaining the rationale of using Low and Middle Income Countries instead of Resource-Limited Settings. The terminology has be amended as requested ( see line 73-74). 

Reviewer’s comment: 

2. The other reviewer has complained about too many abbreviations, and there are still too many abbreviations / acronyms. I would limit the paper to 3-5 acronyms and then spell out everything else. The reader cannot keep track of all of the acronyms.

Response: The abbreviations have been reduced to 5. The following are the abbreviations in the text, IVD, WHO, COSECA, MCAZ, and ISO. 

Reviewer’s comment

3. The introduction is still significantly too long. The authors need to express the same content, but use half as many words. There is a lot of redundant information in the introduction.

Response: The introduction was reduced by removing the context of the regulatory system in the United States of America and European Union, Convergence, Reliance and Recognition Mechanisms, and from the Southern African Development Community region without granular data for Zimbabwe, but aggregated data without identifying the individual countries was excluded. 

Reviewer’s comment

4. In general, the paper is about a very important topic and should be published. However, the authors need significant help from an experienced author to restructure the paper to make it more readable. In its current form, it is too repetitive in some areas, and then does not give adequate details in the methods and results sections. I would encourage the authors to seek out a global health partner, who has more experience publishing scientific manuscripts to help them with the structure of the paper.

Response: The manuscript was reviewed by two reputable researchers, as suggested. Their comments and recommendations were considered for the amended submission that was submitted. The definitions for regulatory functions were listed in a table instead of bullet form. Additionally, a list of indicators presented in bullet form was converted into a table, including the description of each indicator. Redundant information in the introduction was removed. The discussion was reduced to avoid repeating the findings reported in the results section. In this section, we compared our findings with other studies that have been conducted. We also indicated new findings and the strengths of our findings.

---

## [Decision Letter · Decision Letter 2]

14 Feb 2024

PONE-D-23-17232R2Assessing Operational Readiness: Regulatory Landscape and Compliance in Zimbabwe for Medical Devices and In Vitro Diagnostic Medical DevicesPLOS ONE

Dear Dr. Chiku,

Thank you for submitting your manuscript to PLOS ONE. After careful consideration, we feel that it has merit but does not fully meet PLOS ONE’s publication criteria as it currently stands. Therefore, we invite you to submit a revised version of the manuscript that addresses the points raised during the review process.

We look forward to receiving your revised manuscript.

Kind regards,

Zenewton André da Silva Gama, Ph.D.

Academic Editor

PLOS ONE

Journal Requirements:

**Additional Editor Comments:**

The reviewer comments on an important requirement. Please consider reducing the size of the discussion to avoid any repetition with the results section. The discussion should focus on: contributions of the study to the area, why you had these results, what are the consequences of your findings, how this relates to other published studies, limitations and future studies. 

Reviewers' comments:

Reviewer's Responses to Questions

**Comments to the Author**

1. If the authors have adequately addressed your comments raised in a previous round of review and you feel that this manuscript is now acceptable for publication, you may indicate that here to bypass the “Comments to the Author” section, enter your conflict of interest statement in the “Confidential to Editor” section, and submit your "Accept" recommendation.

Reviewer #2: (No Response)

2. Is the manuscript technically sound, and do the data support the conclusions?

Reviewer #2: Yes

3. Has the statistical analysis been performed appropriately and rigorously? 

Reviewer #2: N/A

4. Have the authors made all data underlying the findings in their manuscript fully available?

Reviewer #2: Yes

5. Is the manuscript presented in an intelligible fashion and written in standard English?

Reviewer #2: Yes

6. Review Comments to the Author

Reviewer #2: This manuscript is greatly improved from the last revision, and it is almost there!

I think the only remaining is that the discussion section is too long, and they are basically re-reporting the results. They need to significantly reduce the discussion section. Instead of restating the results, the authors should just describe how this study fits in with work that has previously need done, and maybe give some details regarding why the regulatory system is so underdeveloped and if there are any plans or what is needed to change this in the future.

7. PLOS authors have the option to publish the peer review history of their article (what does this mean?). If published, this will include your full peer review and any attached files.

Reviewer #2: No

---

## [Author Response · Author response to Decision Letter 2]

7 Mar 2024

Reviewer’s comment: Please review your reference list to ensure that it is complete and correct. If you have cited papers that have been retracted, please include the rationale for doing so in the manuscript text, or remove these references and replace them with relevant current references. Any changes to the reference list should be mentioned in the rebuttal letter that accompanies your revised manuscript. If you need to cite a retracted article, indicate the article’s retracted status in the References list and also include a citation and full reference for the retraction notice.

Response: The reference list was reviewed. We verified if any of the cited sources were retracted by accessing external websites or databases for the retraction status of references. We searched the references from PubMed, Google Scholar, and the respective publishers' websites, namely the World Health Organization, Global Harmonization Task Force (GHTF), International Medical Devices Regulators Forum (IMDRF), and Regulatory Professionals Affairs Society (RAPS). We looked for any retraction notices or updates. We noted that the web links for references 1, 10, 13,14, 15 and 16 had been changed from the source. We amended the web links with the new links. Additionally, the WHO website was intermittently accessible, which may have made the references from the WHO website inaccessible. 

Additionally, reference number 23 was added as a supplement to reference 22. Reference 23 is an addendum to reference 22. 

Review Comments to the Author

Reviewer #2: This manuscript is greatly improved from the last revision, and it is almost there!

I think the only remaining is that the discussion section is too long, and they are basically re-reporting the results. They need to significantly reduce the discussion section. Instead of restating the results, the authors should just describe how this study fits in with work that has previously need done, and maybe give some details regarding why the regulatory system is so underdeveloped and if there are any plans or what is needed to change this in the future.

Response: The discussion was reviewed to remove the results. There was not much literature specific to the Zimbabwean setting; there was only one study by Hubner et al.( https://pubmed.ncbi.nlm.nih.gov/33764886/ ) that had granular data specific to Zimbabwe to compare with. Other studies by Mwedzi-Dube et al.( https://pubmed.ncbi.nlm.nih.gov/33029353/ ) and Kniazkov et al. (https://pubmed.ncbi.nlm.nih.gov/33088577/) were for Southern Africa, including Zimbabwe. However, the data was not granular enough to identify where Zimbabwe stood, and the studies were not specific for regulating medical devices and IVDs but health products in general. Therefore, the introduction of the manuscript starts with the statement, “ This is the first study of Zimbabwe’s medical devices and IVD regulations, focussing on all regulatory functions.” (Lines 286-287 of Revised Manuscript with Tracked Changes). 

The reason why the regulatory system is underdeveloped is stated in lines 287 -294 as follows, “The regulatory system concerning medical devices and IVDs in Zimbabwe is notably deficient due to an underdeveloped legislative framework for regulatory functions. The Medicines and Allied Substances Control Act was enacted in 1969 and only focused on the medicine framework. Subsequent amendments were implemented to include the formulation and implementation of regulation of condoms and gloves. The amendments did not cover all the regulatory functions for an effective regulatory system proportionate to the different health products. This is why the regulatory system for medical devices and IVDs is underdeveloped.”

The authors suggested what needs to change based on the findings. Lines 506-540 of the Revised Manuscript with Tracked changes propose changes in the Medicines and Allied Substances Control Act to incorporate the crucial components of the legislative framework as follows, 

“To effectively regulate medical devices and IVDs in Zimbabwe, The Medicines and Allied Substances Control Act needs to be amended to incorporate key components into the legislative framework:

1. Definition and Classification: Clearly define medical devices and IVDs, establishing a risk-based classification system for regulatory oversight.

2. Regulatory Authority Establishment: Empower the MCAZ with adequate resources and expertise to enforce regulatory requirements effectively.

3. Licensing and Registration: Outline requirements for the licensing or registration of manufacturers, importers, and distributors, ensuring compliance with quality management systems and post-market surveillance obligations.

4. Quality Management Systems: Mandate implementation of quality management systems by manufacturers, adhering to specified standards and guidelines, with regular audits and inspections for compliance verification.

5. Conformity Assessment: Establish procedures for conformity assessment to ensure devices meet regulatory requirements, including pre-market approval, product testing, and clinical evaluations.

6. Post-Market Surveillance: Mandate post-market surveillance systems for monitoring device safety and performance, with provisions for adverse event reporting, recalls, and periodic re-evaluations.

7. Labeling and Instructions for Use: Require clear labeling and instructions for use to ensure safe and effective device utilization by healthcare professionals and end-users.

8. Capacity Building and Training: Include provisions for capacity building and training initiatives to enhance regulatory capacity among stakeholders, including regulatory authorities, healthcare professionals, and industry representatives. The competence framework must be put in place that includes proficiency in scientific and health concepts, encompassing the understanding and applying evolving basic and translational science, regulatory science, and public health principles to enhance healthcare product development, evaluation, and oversight (21–23).

9. International Harmonization: Align legislative provisions with international standards and guidelines to facilitate trade and harmonization efforts, potentially adopting regulations developed by the International Medical Devices Regulators Forum and WHO.

Incorporating these components into the legislative framework enables Zimbabwe to establish a robust regulatory system for medical devices and IVDs, ensuring their safety, quality, and effectiveness in healthcare delivery.”

---

## [Editor Report · Decision Letter 3]

5 Apr 2024

Assessing Operational Readiness: Regulatory Landscape and Compliance in Zimbabwe for Medical Devices and In Vitro Diagnostic Medical Devices

PONE-D-23-17232R3

Dear Dr. Chiku,

We’re pleased to inform you that your manuscript has been judged scientifically suitable for publication and will be formally accepted for publication once it meets all outstanding technical requirements.

Kind regards,

Zenewton André da Silva Gama, Ph.D.

Academic Editor

PLOS ONE

---

## [Editor Report · Acceptance letter]

2 May 2024

PONE-D-23-17232R3 

PLOS ONE

Dear Dr. Chiku, 

I'm pleased to inform you that your manuscript has been deemed suitable for publication in PLOS ONE. Congratulations! Your manuscript is now being handed over to our production team.

Kind regards, 

on behalf of

Prof. Dr. Zenewton André da Silva Gama 

Academic Editor

PLOS ONE